# Impact of COVID-19 on healthcare utilization, cases, and deaths of citizens and displaced Venezuelans in Colombia: Complementary comprehensive and safety-net systems under Colombia's constitutional commitment

Donald S. Shepard[1]*, Adelaida Boada[2], Douglas Newball-Ramirez[2], Anna G. Sombrio[1], Carlos William Rincon Perez[2], Priya Agarwal-Harding[1], Jamie S. Jason[1], Arturo Harker Roa[2], Diana M. Bowser[1]

1 The Heller School of Social Policy and Management, Brandeis University, Waltham, Massachusetts, United States of America, 2 School of Government, Universidad de los Andes, Bogotá, Colombia

* shepard@brandeis.edu

## Abstract

### Objective

Colombia hosts 1.8 million displaced Venezuelans, the second highest number of displaced persons globally. Colombia's constitution entitles all residents, including migrants, to life-saving health care, but actual performance data are rare. This study assessed Colombia's COVID-era achievements.

### Methods

We compared utilization of comprehensive (primarily consultations) and safety-net (primarily hospitalization) services, COVID-19 case rates, and mortality between Colombian citizens and Venezuelans in Colombia across 60 municipalities (local governments). We employed ratios, log transformations, correlations, and regressions using national databases for population, health services, disease surveillance, and deaths. We analyzed March through November 2020 (during COVID-19) and the corresponding months in 2019 (pre-COVID-19).

### Results

Compared to Venezuelans, Colombians used vastly more comprehensive services than Venezuelans (608% more consultations), in part due to their 25-fold higher enrollment rates in contributory insurance. For safety-net services, however, the gap in utilization was smaller and narrowed. From 2019 to 2020, Colombians' hospitalization rate per person declined by 37% compared to Venezuelans' 24%. In 2020, Colombians had only moderately (55%) more hospitalizations per person than Venezuelans. In 2020, rates by municipality between Colombians and Venezuelans were positively correlated for consultations (r =

**Data Availability Statement:** Epidemiologic surveillance and mortality data are available from SIVIGILA, National Institute of Health, Colombia (https://www.ins.gov.co/Direcciones/Vigilancia/Paginas/SIVIGILA.aspx). Population data by nationality and municipality are available from the Department Administrative Nacional Estastica (DANE), Colombia, Population by Municipality and Age, Annex on population projections by municipality with simple ages, 2018-2023 (Anexos-proyecciones-poblacion-municipios-edadessimples-2018-2023), cited January 1, 2022 (https://www.dane.gov.co/index.php/estadisticas-por-tema/demografia-y-poblacion/proyecciones-de-poblacion). Health services utilization by municipality and nationality data are available from Registros Individuales de Prestación de Servicios de Salud – RIPS, (https://www.minsalud.gov.co/proteccionsocial/Paginas/rips.aspx). The list of municipalities and additional information are presented in the Supporting information file.

**Funding:** This study was funded primarily by the project "Strengthening the humanitarian response to COVID-19 in Colombia" through a grant to Brandeis University from Elrha's Research for Health in Humanitarian Crises (R2HC) Program (elrha.org) with DMB as PI, which supported all authors. R2HC is funded by the UK Foreign, Commonwealth and Development Office, Wellcome, and the Department of Health and Social Care through the National Institute for Health Research. The remaining funding was provided by the World Bank (worldbank.org) through award WORLD 8006362 to Columbia University, Sub-award 2(PG006778-01) to Brandeis University (DSS, Brandeis PI) for the project "The big questions on forced migration in health." It supported DSS, PA-H, JSS, AHR, and DMB. The funders had no role in study design, data collection and analysis, decision to publish or preparation of the manuscript. There was no additional external funding received for this study.

**Competing interests:** The authors have declared that no competing interests exist.

0.28, p = 0.04) but uncorrelated for hospitalizations (r = 0.10, p = 0.46). From 2019 to 2020, Colombians' age-adjusted mortality rate rose by 26% while Venezuelans' rate fell by 11%, strengthening Venezuelans' mortality advantage to 14.5-fold.

## Conclusions

The contrasting patterns between comprehensive and safety net services suggest that the complementary systems behaved independently. Venezuelans' lower 2019 mortality rate likely reflects the healthy migrant effect (selective migration) and Colombia's safety net healthcare system providing Venezuelans with reasonable access to life-saving treatment. However, in 2020, Venezuelans still faced large gaps in utilization of comprehensive services. Colombia's 2021 authorization of 10-year residence to most Venezuelans is encouraging, but additional policy changes are recommended to further integrate Venezuelans into the Colombian health care system.

## Introduction

COVID-19 has challenged health systems in almost every country, but particularly those in low- and middle-income countries [1]. The Global Fund for AIDS, Malaria and Tuberculosis received responses from 106 participating countries about the pandemic. Respondents (which include Colombians), reported that the pandemic had interrupted the treatment and diagnosis of all three of the Fund's target diseases. With increased service demands due to the pandemic, health systems in low- and middle-income countries reported shortages of medical supplies, equipment, and qualified workers, and disruptions in service delivery. These problems limited the capacity of their systems to respond to non-COVID concerns [2].

Countries receiving many forcibly displaced people face compounded challenges from added demands. Their host populations may face overcrowded facilities while displaced populations are unable to access specialized care [3]. Of the 82.4 million forcibly displaced persons worldwide as of 2021 [4] 39% lived in just five countries. These countries were distributed across the World Bank's income categories: Germany (high), Colombia and Turkey (upper-middle), Pakistan (lower-middle), and Uganda (low) [4]. All these countries face the simultaneous challenges of serving both host and displaced populations for healthcare, social services, shelter and security. Even before the COVID-19 pandemic, discrimination, lack of information, high costs, and fear of deportation created barriers for displaced populations to accessing health care [5] or other services [6]. The pandemic exacerbated these problems [7].

Under Colombia's constitution, all persons in the country are entitled to limited lifesaving medical care. In addition, citizens and officially registered foreigners are entitled to register for health insurance under the country's national health system. The International Monetary Fund estimated that investing in Venezuelan migrants' services could boost Colombia's gross domestic product (GDP) by up to 4.5% by 2030 [8]. Nevertheless, migrants' health needs often exceed the capacity of the existing health and social support systems [9].

Colombia's healthcare funding mechanisms, described further below, have created two complementary components of Colombia's healthcare system. Life-saving or safety net services, which are largely hospital services, are available to all persons, while comprehensive services, which are largely consultations, are available only to persons enrolled in an insurance plan. Consultations are formal contacts with a medical professional, such as a physician, for

diagnosis and possible treatment in a non-emergency, ambulatory setting. For Venezuelans, enrollment required authorization, documentation, and mastery of the necessary administrative steps [10–12].

The objective of this study is to assess Colombia's achievements in providing healthcare to Venezuelan migrants in comparison to the country's care for its own citizens during the first year of the COVID-19 pandemic. This assessment compared both inputs (such as rates of comprehensive and life-saving services provided) and outputs (COVID-19 case rates and death rates). Some parts of this manuscript were included in the authors' knowledge brief [12].

## Materials and methods

### Conceptual framework

We conceptualized each person's options concerning health services utilization as a balance of opposing factors, as shown in Fig 1. These factors can operate at the level of the municipality (M), the individual (I), or both (M & I). Access and quality of healthcare services, insurance coverage, medical need, patients' awareness of their rights, and patients' insurance coverage all promote utilization. Barriers to services include lockdown restrictions and facility staffing shortages. Municipal-level factors should affect Venezuelans and Colombians similarly, individual-level factors may differ both between and within nationalities.

On top of pre-existing barriers, the COVID-19 pandemic placed several barriers on accessing services. First, the general lockdowns that began in March 2020 made transport less accessible. While consultations with physicians were allowed movements, patients may have feared infection while visiting or traveling to and from a health facility. Given the concern about minimizing risks to medical personnel, the health system explicitly focused on emergency care but reduced the supply of non-emergency services. For example, some hospital staff shifted to remote and part-time work (often 3 of the 5 workdays). Also, protective measures in healthcare facilities to space patients apart lowered the number of services that could be delivered in a workday, thereby decreasing patients' ability to obtain in-person appointments.

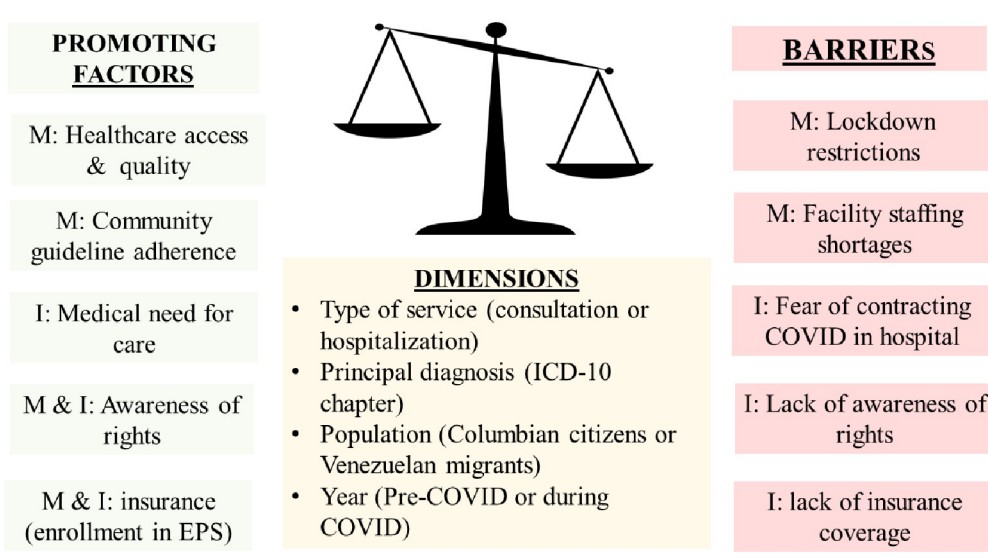

**Fig 1. Conceptual framework: Utilization of medical services in Colombia: A balance.** Notes: M denotes municipality; I denotes individual; ICD denotes International Classification of Diseases; COVID denotes corona virus disease.

## Setting

**Venezuelan migration to Colombia.** Colombia hosts around 1.8 million displaced Venezuelans, which is the second-largest number of migrants in any host country globally [4, 13]. Most migrants, who began arriving in 2015 primarily to escape Venezuela's economic collapse, have located in large cities such as Bogotá, Barranquilla, Cúcuta, and Medellín, searching for relative safety and work opportunities [14].

This study is part of a family of studies examining Colombia's health system performance for citizens and displaced Venezuelans [10–12, 15, 16]. A previous publication from this family of studies, a telephone survey of 8,130 Venezuelan migrants and Colombian nationals, focused on the most frequently used health service, consultations [10]. By using Colombia's extensive administrative data, the present study also examined less frequent services, such as hospitalizations.

Whereas some countries have been unwelcoming by not allowing refugees the right to access humanitarian services, such as collectively funded health care and legal employment, Colombia has been relatively hospitable to displaced Venezuelans. From the beginning of the most recent migratory crisis through January 2021, almost half of displaced Venezuelans already had the opportunity to register, work legally, and receive benefits of temporary residency. Analysts trace this line of policy to the 1990s, when Colombia's internal armed conflict and illicit drug violence forced millions of Colombians to emigrate—many to Venezuela [17]. Colombians have been grateful for past assistance from Venezuela when Colombia faced domestic violence [18].

**Colombia's insurance regimes.** Persons in Colombia have three approaches for publicly funded personal health services: (1) Most formal sector workers and their families are members of the "contributory regime" of an *Entidad Promotora de Salud* (EPS), a health management organization, funded by the workers and their employers through a payroll tax. (2) Most other citizens and officially registered Venezuelans and other foreigners, such as informal workers, belong to an EPS under the "subsidized regime," where premiums are paid by the national government's general revenues. Although the law indicates that both contributory and subsidized members theoretically access identical Health Benefits Plans, subsidized members tend to have greater needs but often less access to some services (e.g. easily filling prescriptions) [19, 20]. (3) Even if not enrolled in an EPS or registered, any person in Colombia (citizen or migrant) is entitled to lifesaving services. However unenrolled persons are entitled only to emergency care and mass public health services [21].

## Data sources

**Overview.** The study combines three powerful databases analyzed at the municipality level: (1) disease surveillance and deaths, *Sistema Nacional de Vigilancia en Salud Pública* (SIVIGILA) [22], (2) health care utilization, the *Registro Individual de Prestación de Servicios* (RIPS) [23], and (3) the national census from DANE [24]. Using these databases, we analyzed access to health care for displaced Venezuelans and Colombians before and during the COVID-19 pandemic on rates of hospitalization, consultation, and COVID-19 cases, and deaths. With the paper's goal of contrasting comprehensive versus safety net services, we focused on the types of care to best fit those paradigms. We therefore selected consultations, rather than emergency room visits, as our utilization measure for comprehensive care and hospitalizations as our indicator for safety net services. A unique feature is the study's paired municipal-level analyses across the 60 Colombian municipalities with the largest numbers of Venezuelan migrants, comparing years, indicators, and nationalities in the same city (S1 File).

**Health services utilization.** Our main data source for the quantities of health care services received by displaced Venezuelan and Colombian citizens by municipality of residence and month of service was the RIPS. It records all healthcare system transactions in Colombia through the country's system of universal health insurance. The transactions, including diagnoses [25], were transmitted from the institution that provides the service, termed an *Institución Prestadora de Servicios* (IPS), such as a hospital or clinic, through the member's EPS to the central registry (RIPS). These data on service utilization serve as the numerators for municipality-based rates of hospitalizations and consultations. Although RIPS also reports numbers of procedures, we did not count them to avoid overlap with our main measures. The EPS's are also the funders of the IPS's on behalf of the EPS members. Since the system is used to pay facilities, the usual reporting lag was only 1–2 months. The utilization data for this study are based on the early part of the pandemic (March 1 through July 31, 2020) and the same dates for 2019. The numbers shown here are period-based rates based on aggregate numbers of events, rather than person-level analyses. Thus, RIPS could include more than one consultation or hospitalization for the same person. Residents of Colombia or their employers can also purchase private health insurance and/or pay out of pocket for services from private providers.

**COVID-19 cases and deaths and overall deaths.** SIVIGILA, operated by the National Health Institute and the Ministry of Health and Social Protection, reports officially confirmed COVID-19 cases and fatalities by municipality and week. In addition, it reports overall deaths by month by municipality, and age breakdowns for Colombians.

**Denominators.** The denominators are the number of persons in the country and each study municipality by nationality and age. These denominators came from the Colombian census for Colombian citizens [24] and from the border control agency for displaced Venezuelans [26, 27]. The proxy measure of enrollment in contributory insurance in each municipality was derived from RIPS. The contributory regime's share was proxied as the proportion of health services delivered through the contributory regime. It was calculated as the municipality's sum of services (ambulatory, inpatient, and procedures) provided to contributory EPS enrollees divided by the municipality's grand total of all services.

## Analytical approach

**COVID-19 events.** We derived weekly rates of COVID-19 cases, deaths, and period rates of hospitalizations and consultations by dividing the aggregate counts during the study periods by the corresponding populations and expressed them as rates per 100,000 population. To accommodate potential wide ranges of rates among different indicators, we expressed all values on common logarithmic (base-10) scales. We chose base-10, rather than natural logarithms, to facilitate interpretation. As about half the municipalities had no Venezuelan COVID-19 deaths, we could not perform the logarithmic transformation with the original data as the log of zero is undefined. Instead, for COVID-19 deaths we used a "shifted" log scale in which the constant 1 (i.e., 1 death) was added to the count of Colombian and Venezuelan deaths in each municipality before taking the logarithm.

For COVID-19 cases, where data were tallied by epidemiologic weeks, the study period was March 1, 2020 through November 28, 2020. However, due to the lag initiating reporting of deaths by nationality, breakdown of deaths covers April 26, 2020 through November 28, 2020. As the same dates were used for all municipalities, the differences in dates between cases and deaths adjusted for the lag between diagnosis and death, and did not affect relative rates among municipalities.

**Overall deaths.** To examine overall death rates to most closely approximate the COVID-19 rates, we first calculated the age-standardized mortality rates in May through November 2019 and the corresponding months for 2020 for Colombians and Venezuelans for each of the 60 selected municipalities. These 7-month periods created comparability between COVID-19 and overall death rates. To control for changes in the age over time as well as differences between the two nationalities, all of our analyses used the Colombians' 2019 age distribution.

Age standardization was based on three age categories: 0–49 years, 50–69 years, and 70 + years, with population data from DANE [24] In 2019, the respective national population totals and age shares by nationality were 49,395,678 and 75.2%, 18.3% and 6.5% for Colombian citizens and 1,771,237 and 92.6%, 6.7% and 0.8% for Venezuelans, respectively. The 2019 shares of deaths by age group, which combined both nationalities, were 21.3%, 22.4% and 56.3%, respectively [28]. Age standardization corrected for the displaced Venezuelans being substantially younger than the Colombians. We then adjusted for differences between the data sources for age- and municipality-breakdowns and between the 60 municipalities and Colombia overall by dividing both Colombians' and Venezuelans' age-standardized mortality by 1.88. The adjusted age-standardized rate for Colombians was then 1.00.

**Health services utilization.** Rates of all-cause hospitalizations and consultations per 100,000 population by municipality were analyzed for Colombians and Venezuelans for identical time periods in 2019 and 2020, and by time segments within 2020. Comparisons between the time segments of 2020 checked whether the relationships were enduring as Colombia's response to the COVID-19 pandemic evolved. Correlations of utilization by Venezuelans with that of Colombians examined whether there were common features in a municipality's health system, such as the number, locations, and quality of its health facilities, which affected Colombians and Venezuelans in similar ways. Regressions based on city size further examined one proxy for the sophistication of a municipality's health system and other support systems. Correlations of hospitalizations against consultations for each nationality examined the extent to which consultations (an indicator of ambulatory care) served as a gateway to hospitalizations by identifying health problems requiring further care, or as a substitute through early intervention, averting the need for later inpatient care. Breakdowns of changes from 2019 to 2020 by diagnosis (using aggregate national data) showed how the pandemic differentially affected health problems of varying degrees of urgency.

**Impact of insurance coverage.** The aforementioned household survey found that 97.0% of Colombians across 60 municipalities reported being enrolled in an EPS (a key measure of insurance coverage) compared to only 26.4% of Venezuelans. However, after controlling for self-reported EPS enrollment, the analysis found that Colombians' and Venezuelans' self-reported access to health services was comparable [10]. We wished to examine this relationship further with RIPS data. To do this, we needed to be able to analyze small numeric differences near the extremes of a proportion (i.e., 0 and 1), so we transformed the proportions using the logit scale. The logit scale is defined as the natural logarithm of p/(1-p), where p is the proportion of services for the specified municipality for a given nationality.

## Ethical exemption

The protocol (#21010R-E Bowser) for the family of studies encompassing this work was deemed to be exempt in accordance with 45 CFR 46.104(d) 2 by the Brandeis University Human Research Protection Program (Institutional Research Board) based in the United States on Sept. 17, 2020.

## Results

### COVID-19 case and death rates

Fig 2A (i.e., Fig 2, panel A) shows the Venezuelan versus Colombian COVID-19 case rates in 2020, with each municipality represented by a dot. The horizontal (x) axis shows each municipality's rate for Colombians while the vertical (y) axis shows the same municipality's rate for Venezuelans. The dashed 45-degree line here and in subsequent figures shows the line of equality that would apply if a municipality had identical x and y values. On average, Colombians have substantially higher reported COVID-19 case rates than Venezuelans by an average factor of 10.64. Venezuelans' reported COVID-19 rate in each municipality was highly correlated with that of the municipality's Colombian residents (r = 0.52, p<0.0001). When the 2020 study period was subdivided into two equal time segments, the Colombian rate proved consistently higher in both periods (S2 File).

In addition, masking, social distancing, and access to testing likely varied among municipalities. The amount of variation in COVID-19 case rates among municipalities was comparably large for the two nationalities, as evidenced by similar multiples for the interquartile ranges (third quartile as a multiple of the first quartile of 3.75 for Colombians and for 4.35 for Venezuelans).

Fig 2B shows the relationship between Colombian and Venezuelan COVID-19 average weekly death rates by municipality. The actual (not transformed) COVID-19 population-weighted average weekly death rates of Colombians and Venezuelans across these municipalities were 2.57 and 0.56 per 100,000 population, respectively. Similar to the pattern for COVID-19 case rates, the death rates of Colombians were significantly higher than those of Venezuelans. However, the relative rate of Colombians compared to Venezuelans for death rates (4.57) was substantially smaller than that for reported case rates (10.67).

The case-fatality rates based on reported cses (approximated by dividing its weekly COVID-19 death rate by its weekly case rate) were 3.63% for Colombians versus 8.29% for Venezuelans. The case-fatality rate of Colombians was only 0.44 times that of Venezuelans. As Colombians are older, on average, than Venezuelans, their case-fatality rate was expected to have been higher than that of Venezuelans if they had had the same access to care, the opposite of what we observed.

Unlike the significant positive correlation for COVID-19 case rates between Colombians and Venezuelans, there was zero correlation between the death rates of the two populations (r = 0.00, p = 1.00). However, there was substantial chance variation in Venezuelan deaths at the municipal level due to small numbers. The median municipality had only 6,462 Venezuelan residents and only 1 Venezuelan COVID-19 death over the 32-week period. And, as noted, almost half of the municipalities had zero reported Venezuelan COVID-19 deaths. This result is consistent with the interpretation that due to small numbers, the variation in Venezuelans' COVID-19 death rates across municipalities was due to chance.

### Overall death rates

The mean crude all-cause mortality rates per 100,000 population over the 7-month observation periods were 425.5 for Colombians and 41.2 for Venezuelans in 2019 and 530.1 and 36.5, respectively, in 2020. The corresponding age-standardized mortality rates (standardized to the Colombians' age distribution) were 425.5 and 94.2, respectively, in 2019 and 535.8 and 83.4, respectively, in 2020. Because the Venezuelans are so much younger than the Colombians, their age-standardized death rates are about twice their crude rates.

Fig 3A plots the adjusted 2020 age-standardized relative mortality of Venezuelans against that of Colombians in the same municipality. The labeled cities are the five with the most

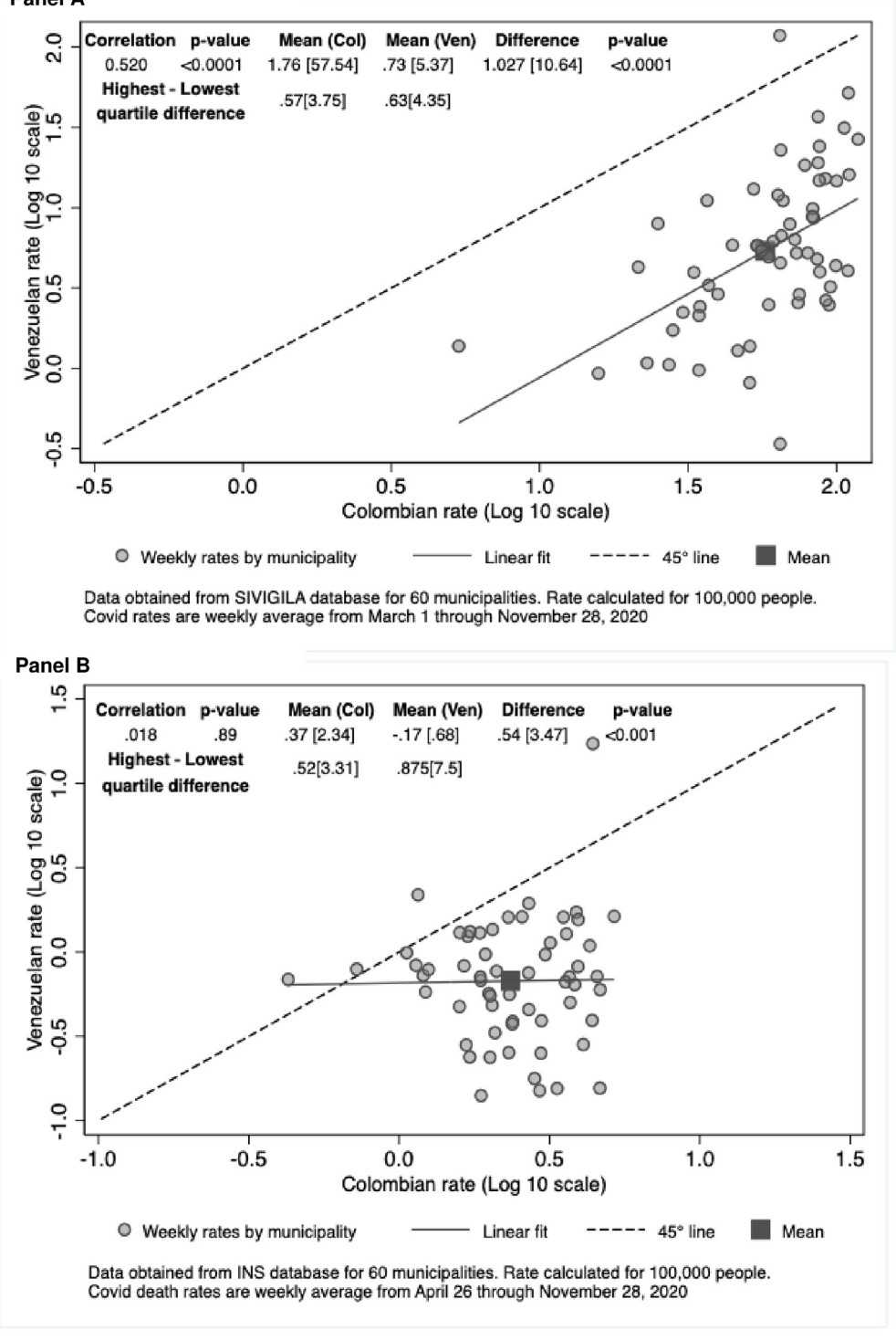

**Fig 2. Venezuelan versus Colombian COVID-19 rates per 100,000 population in 2020 by municipality in (A) cases and (B) deaths.**

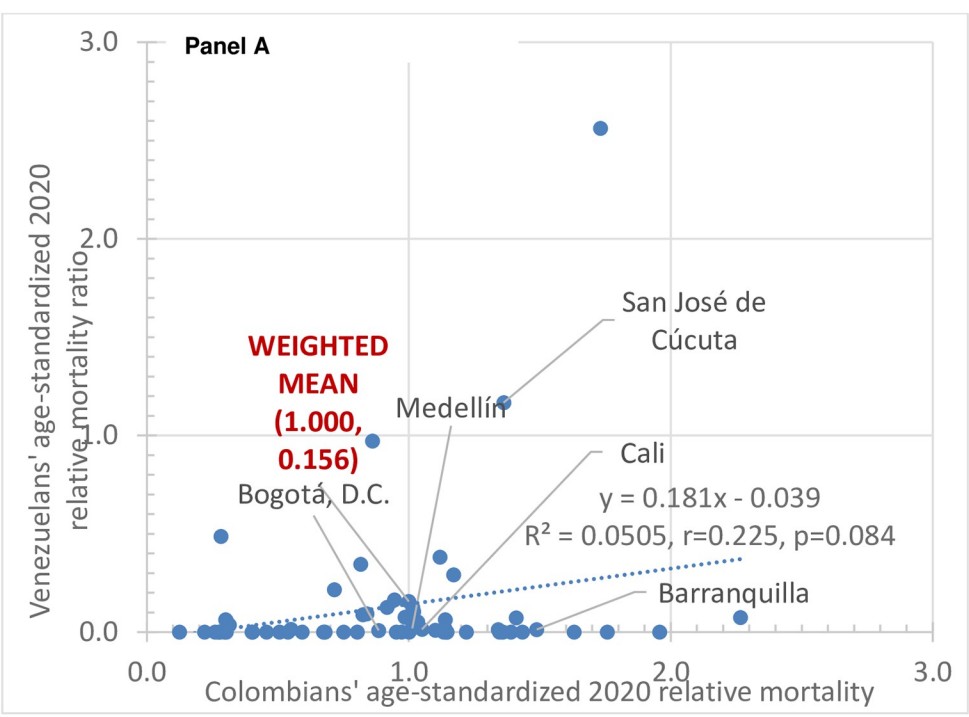

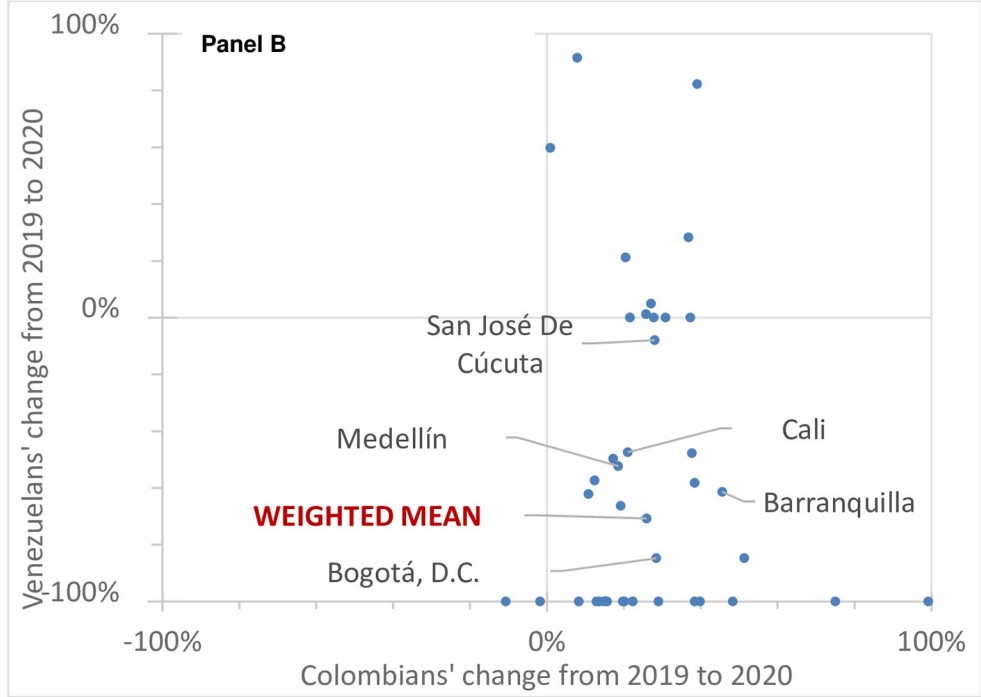

**Fig 3. Adjusted age- standardized relative mortality by municipality and nationality showing (A) levels in 2020 and (B) percentage change from 2019 to 2020.**

displaced Venezuelans in 2019. The correlation is positive, as expected, with borderline statistical significance (p = 0.084). This result supports the fact that some municipal-level factors, such as access to and quality of the health system, and environmental and social determinants of health, affect both nationalities. As with COVID-19 deaths, many municipalities had no Venezuelan deaths due to their small numbers of Venezuelans and their young ages.

Fig 3B plots the change in the-age-standardized mortality for Venezuelans against the change for Colombians. The weighted mean changes are plus 25.9% for Colombians (a moderate increase) and minus 11.4% for Venezuelans (a modest decline). The adjusted age-standardized death rates of Venezuelans were 78% below the corresponding Colombians' rates in 2019 and 84% below in 2020.

Municipal-level calculations of change were possible for 38 municipalities, of which 28 (74%) showed an increased rate for Colombians but a decrease for Venezuelans. The change could not be calculated for the remaining 22 municipalities because they reported zero Venezuelan deaths in the 2019 study period, so the mortality ratio was undefined for Venezuelans. In addition, for one municipality, the change (+150%) exceeded the maximum value (100%) in Fig 3B.

## Overall hospitalization and consultation rates

As the COVID-19 pandemic reduced staffing at health facilities and created travel barriers, it reduced access. Fig 4A shows overall (all-cause) hospitalization rates by municipality in 2020 versus 2019. Hospitalization rates fell modestly for both nationalities: specifically, by 37% (i.e., 1.00 minus 0.63) for Colombians and 24% (i.e., 1.00 minus 0.76) for Venezuelans. These were likely the result of lockdowns and fear of visiting a hospital. The gap between the nationalities in hospitalization rates per 100,000 population narrowed from 2019 to 2020. In 2019, Venezuelans were 45% below Colombians (calculated as 1-1/1.82). By 2020, the gap fell to 35% (calculated as 1-1/1.55).

Fig 4B shows consultation rates by municipality in 2020 versus 2019. Consultation rates fell by slightly more than hospitalizations for both nationalities (by 42% (calculated as 1.00 minus 0.58) for Colombians, and 37% for Venezuelans (calculated as 1.00 minus 0.63), again presumably from lockdowns and fear of visiting a health facility. Colombians had substantially higher rates of consultations compared to Venezuelans in both 2019 (multiplicative factor of 7.76) and 2020 (factor of 7.08). Stated differently, Venezuelans' consultation rates were 87% below those of Colombians in 2019. In 2020, the gap was trimmed slightly to 86% below Colombians. However, each municipality's rate of consultations relative to the grand mean of all municipalities was generally stable for both Colombians and Venezuelans between the two years, as shown by the highly significant positive inter-year correlations (0.21 for Colombians and 0.35 for Venezuelans). Rates varied across municipalities much more widely for Venezuelans (interquartile ranges 12-20-fold) compared to Colombians (interquartile range of 2-fold).

As with hospitalizations, the higher age of Colombians, noted above, had been expected to be an additional explanatory factor for their higher rates of consultations compared to Venezuelans. However, further analysis of a subset of the consultation data found evidence to the contrary. Tabulations of RIPS for March through May 2020 gave age-specific consultation rates for Venezuelans. These were weighted according to age distribution of Colombians to give an age-standardized consultation rate for Venezuelans. While Venezuelans' age standardized rate had been expected to be higher than their crude rate, it turned out to be 0.8% lower. The reason could be that Venezuelans aged 50–69 had a lower age-specific

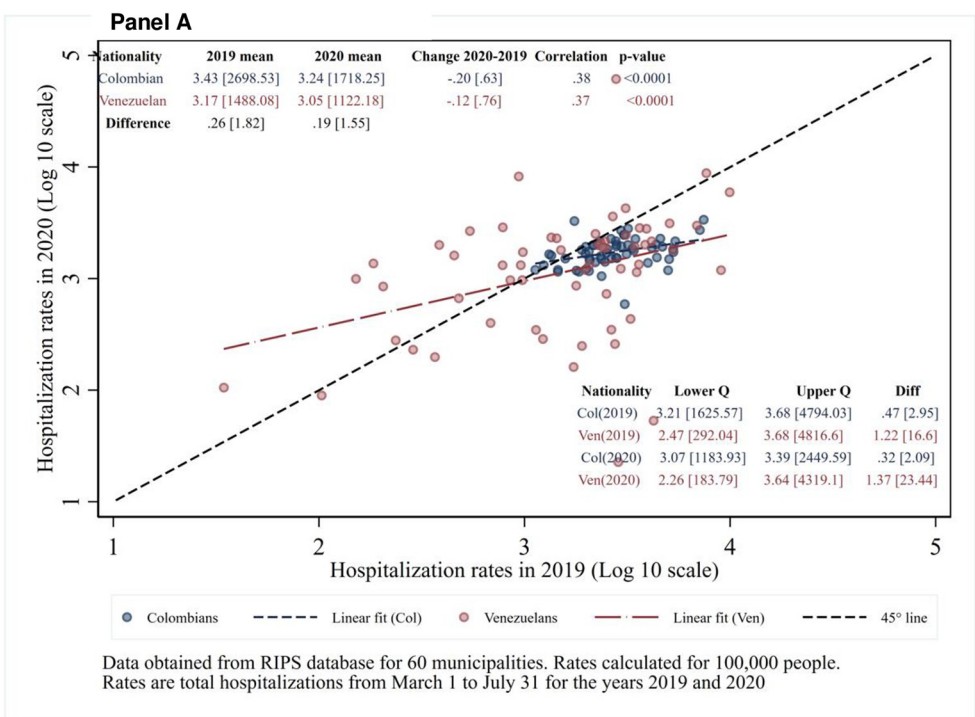

**Fig 4. All-cause rates of Colombians and Venezuelans in 2020 versus 2019 for (A) hospitalizations and (B) consultations.**

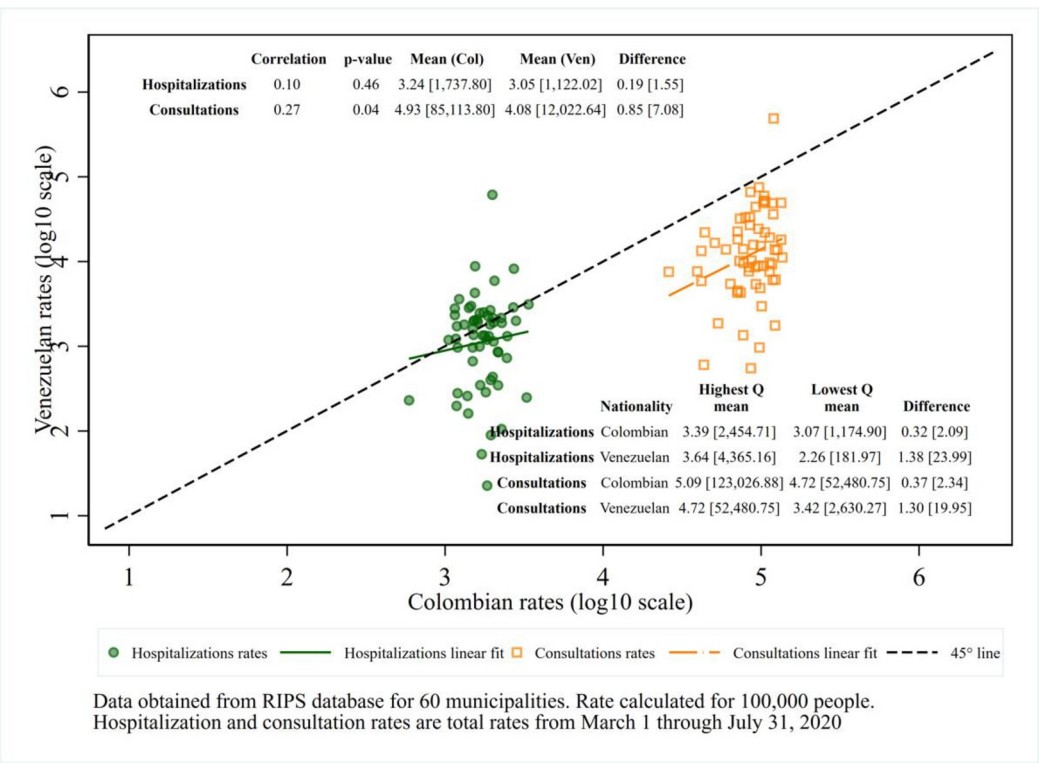

**Fig 5. Hospitalization and consultation rates in 2020: Colombians versus Venezuelans.**

consultations rate than their younger peers, similar to the pattern for hospitalizations, perhaps because they were even more fearful of visiting a healthcare facility and being exposed to COVID-19.

## Relationship between hospitalization rates and consultation rates

Fig 5 compares displaced Venezuelans and Colombians on both hospitalization and consultation rates in 2020: There was a significant positive correlation in consultation rates between Colombians and Venezuelans in 2020 (r = 0.27, p-value <0.05). As expected, Colombians have substantially higher rates of consultations (7.08 times as high) and less variability across municipalities (interquartile ratio 2.29) compared to Venezuelans (interquartile ratio 20.42). The consultation rates of Venezuelans were positively correlated with those of Colombians in the same municipality (r = 0.27, p = 0.04). On the other hand, the correlation of hospitalizations between the two nationalities was small (r = 0.10) and not statistically significant (p = 0.46). The observations straddle the line of equality (represented by the dashed 45-degree line in Fig 5) and the ratio of 1.55 is relatively close to 1.00. However, although Venezuelans were not substantially behind Colombians on average hospitalization rates, the Venezuelans' rates varied substantially across municipalities (interquartile ratio = 23.44).

Two opposing factors (substitution and complementarity) affect the relationship of hospitalizations to consultations within each nationality: The results in Fig 6A and 6B show that the complementarity effect dominated for both nationalities and for both years. With a high degree of statistical significance (p<0.0001), the rate of hospitalizations was positively related to the rate of consultations for both nationalities for both years.

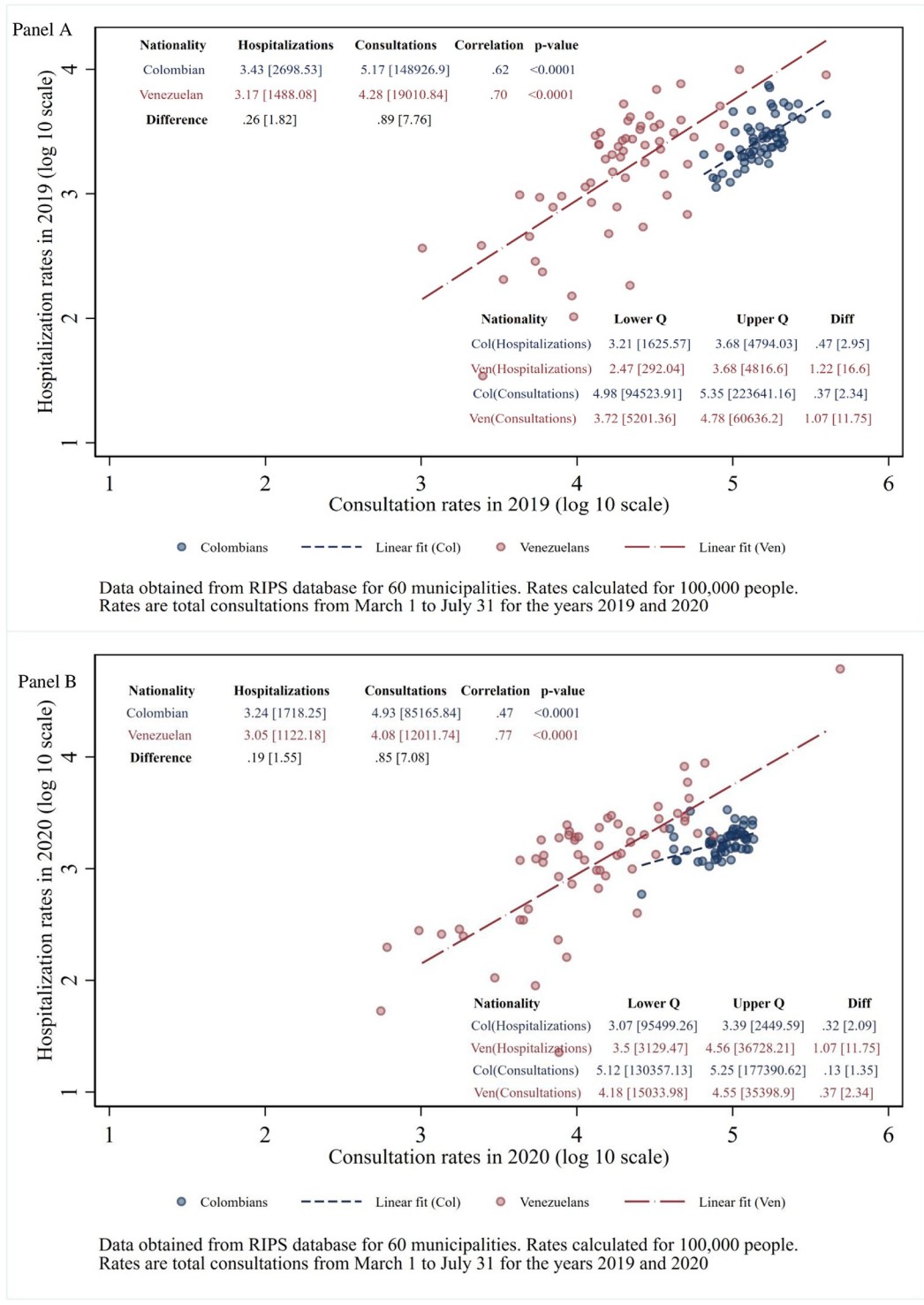

**Fig 6. All-cause hospitalization versus consultation rates for Colombians and Venezuelans by municipality in (A) 2019 and (B) 2020.**

### Analyses of health services utilization by diagnosis, insurance regime and city size

Breakdowns by diagnosis found that the conditions considered more urgent changed less between 2019 and 2020, compared to less urgent needs (S3 File). Findings by insurance regime show that persons in the contributory regime used more services than others, whether Colombian or Venezuelan (S4 File). Breakdowns by city size found that larger cities had higher rates of utilization (S5 File).

## Discussion

### Healthy migrant effect

The most prominent explanation for Venezuelans' consistently lower age-standardized death rates is the "healthy immigrant effect," the tendency of immigrants to be healthier than the corresponding host populations. Researchers generally ascribe this to a selection effect—healthy people desire and are better able to migrate internationally [29, 30].

The fact that most displaced Venezuelans are working-age adults, with proportionately fewer elders than Colombians, suggests that Venezuela's failed economy was the major factor behind their out-migration. The continued decline in Venezuela's per capita GDP from US $2299 in 2019 to $1691 in 2020 may have contributed to intensifying this healthy immigrant effect [31].

### Comparative outcome rates

Disease reporting from the United States Centers for Disease Control and Prevention shows that age-specific hospitalization and death rates per 100,000 population for COVID-19 increase dramatically with age [32]. Cox regression on COVID-19 survival rates in Colombia showed that the hazard rate multiplied five-fold for each higher age category (about 24 years) [33]. In addition to the Colombians' higher ages, the aforementioned healthy immigrant effect, several factors could explain why COVID-19 case and death rates were higher in Colombians and overall deaths rates were higher and rising in Colombians while being lower and declining among Venezuelans. First, the findings are consistent with this study's tenet that Venezuelans had access to Colombia's safety net system before and during the pandemic. Venezuelans continued to access lifesaving care, thereby maintaining their relatively low death rate. Second, if Venezuelans' pre-pandemic use of the healthcare system was mainly for life-threatening problems, their use of the healthcare system would have changed little during the pandemic. Third, by contrast, Colombians, as more discretionary users of their health system, may have been more deterred by fear of exposure to COVID-19 and ended up deferring lifesaving care. Fourth, Venezuelans may have practiced more health-conscious behaviors. As evidence, the study's telephone survey component found that Venezuelans were less likely to report flu-like symptoms related to COVID-19 than Colombians [10]. Similarly, the mobility component of this family of studies found that weekend mobility (when patrons might visit bars and clubs) appeared to be a greater contributor to COVID-19 transmission than mid-week mobility (presumably for work and necessary errands), Venezuelans may have been less mobile due to lower disposable income or less confidence, and thus less exposed [10–12, 15, 16]. Finally, Venezuelans may have become more resilient as they were forced to adapt to the stresses of discrimination, adverse economic conditions, and a foreign environment [29].

More health-conscious behavior by Venezuelans' would be consistent with the economic principle of moral hazard: Venezuelans, knowing that comprehensive care was largely inaccessible, may have worked hard to reduce their risks. Compared to Venezuelans, Colombians in

the lower socio-economic strata (strata 1 through 3) were far likelier to be in the contributory insurance regime, which provided reasonable access to comprehensive care [10]. Thus, they may have taken fewer precautions to avoid illness. Fifth, the vital statistics system may have failed to record some deaths, with greater and widening gaps among Venezuelans.

## Performance of the complementary health system components

Displaced refugee populations have difficulty accessing health services in most parts of the world. While respondents were not broken down by nationality, a survey from other researchers of 1258 primary care patients in Colombia reported lower perceived access to care since before the pandemic [34]. The perceived reductions, which applied to both behavioral and general health, are consistent with the utilization reductions shown in Fig 4B. Under the Colombian constitution, as noted above, all persons in Colombia are entitled to lifesaving medical care. Despite the challenges of the COVID-19 pandemic, during the early COVID-19 pandemic Venezuelans were not too far below Colombians in hospitalization rates per 100,000 population. The gap had narrowed from 45% for the comparable months in 2019 to 35% in 2020. Thus, for health care needs sufficiently serious to require hospitalization, Colombian health care institutions had made relative improvements in meeting legal and medical requirements.

For consultations (part of the comprehensive health system component), however, the gap was wide and narrowed only slightly. According to RIPS, Venezuelans' lower consultation rate per 100,000 population was 87% below that of Colombians in 2019 and 86% below in 2020. We also acknowledge that some Colombians, particularly those in lower socio-economic levels, also faced challenges in accessing healthcare [33]. As COVID-19 testing required regular health system access, the related telephone survey also showed that displaced Venezuelans were statistically significantly less likely to report use of COVID-19 testing compared to their Colombian counterparts [10]. Our analysis sought to distinguish service access from data quality. While the surveillance data used here have not been officially certified, we have no reason to suspect any widespread or systematic reporting errors in relation to cases tested.

The Colombia study was part of a consortium of studies of health under forced displacement across four diverse sites (Bangladesh, Colombia, Democratic Republic of the Congo, and Jordan) [3]. The consortium concluded that Colombia had done the most to integrate its displaced population (Venezuelan migrants) into its national health service. Jordan, hosting Syrian refugees, was next. In all these sites, forced displacement has already existed for several years. Integration is likely to be less expensive and more sustainable over the long run than creating separate humanitarian systems. However, the short run costs can be appreciable, estimated at 0.5% of GDP in Colombia's case [8]. This integration would be facilitated by humanitarian donors assisting host governments with the costs of providing services to displaced populations.

## Recent policy changes

The regularization status for displaced Venezuelans has improved throughout the years, and on February 9, 2021 the Colombian government created the Statute of Temporary Protection for Venezuelan Migrants [35, 36]. The objective of the temporary protection status is to allow displaced Venezuelans living in Colombia to transition from a regime of temporary protection to an ordinary migratory regime. For instance, registered displaced Venezuelans will have 10 years to reside legally in Colombia, enroll in all social protection services, work legally, and eventually transition to a permanent resident visa. This measure seeks to protect the migrant

population that is currently in irregular conditions and reduce current and future irregular migration.

This policy makes Colombia similar to Portugal, Jordan, Qatar, and Ecuador in including displaced populations in its COVID-19 response and providing access to public services for displaced populations [37]. Initially, as migratory regularization and insurance enrollment processes are coordinated, this policy should substantially increase Venezuelans' enrollment in the subsidized insurance regime. Over time, as Venezuelans are hired into formal sector employment, enrollment in contributory schemes should rise.

However, given that enrolling is a multi-step process, public policy makers should anticipate possible challenges and delays in implementation. First, an undocumented Venezuelan must obtain a permit (*Permiso Especial de Permanencia*). This process may have considerable bottlenecks, such as providing legal evidence of residing in the country before January 31, 2021 or having valid identification documents. Then the migrant can apply to enroll in an EPS through local government agencies or the national migration agency (*Migración Colombia*). Another beneficial policy has been the inclusion of all Venezuelans in Colombia in its national vaccination strategy [38].

## Limitations

Three limitations should be noted. First, our health services analyses are based on RIPS. If a health facility does not report a service in RIPS, it does not receive payment. If gaps were widespread and systematic, they could bias this study's results. If such gaps exist, they are likely due to problems within the health facility (e.g., staff do not understand how to enter the information or have technical difficulties). Such omissions, if they occurred, would likely affect both Venezuelans and Colombian patients and both hospitalizations and consultations, but would be limited to just one or two of the municipality's health facilities. Thus, any effect on overall patterns would be small.

Second, validating our observational study was complicated by the fact that it did not fit one of the conventional designs covered by checklists for Strengthening the Reporting of Observational Studies in Epidemiology (STROBE) [39], i.e. cross-sectional, cohort, or case control studies. Our observations were not persons, but rather rates of specified services for each of 2 nationalities for each of 2 years within each of 60 municipalities based on aggregate data. Nevertheless, one of the STROBE checklist's goals, describing the procedures to minimize bias, remained relevant. We addressed this goal in two ways. Several analyses examined a rate for nationality in a municipality in 2020 as a function of the rate for the same nationality by municipality in 2019. Thus, each nationality and municipality serve as its own control. Other analyses examined rates for Venezuelans as a function of the rates for Colombians in the same municipality. While each approach controlled for many potential confounders, limitations remained due to possible changes in a municipality's characteristics between 2019 and 2020, and differences between Colombian and Venezuelan residents in the same municipality. Most notably, we noted above that Colombians tended to be older than Venezuelans. Age standardization reduced, but did not eliminate Colombians' higher death rates, suggesting that there other differences, such as prevalence of co-morbidities and the healthy migrant effect, for which we had not controlled with the data available.

Third, this study was limited to characteristics publicly available at the municipal level. We exploited this capability where possible, such as the aforementioned analysis of utilization by city size (S5 File). This constraint precluded analyzing factors not captured in RIPS linking utilization and person-level characteristics, such as socio-economic status (e.g., social stratum). While further demographic breakdowns would have been theoretically possible,

such breakdowns would have substantially complicated the analysis and likely contributed little, as we did not expect major differences across the municipalities.

## Suggested future policy changes

**Assist Venezuelans in becoming official residents.** Colombia's experience demonstrates that a middle-income country has been able to provide at least emergency medical services to its refugee population within its national health service. The country's announced plan to allow most displaced Venezuelans to register to remain legally for 10 years should increase their integration in the future. As migrants are younger on average than their host counterparts and move between Colombia and Venezuela based on economic conditions, their skills and motivation could help grow the Colombian economy—not stealing jobs but building demand [8]. The Colombian government could expedite this process by training government workers and assisting non-government organizations to help identify those eligible and to assist their official registration.

**Strengthen health promotors.** Colombians' dramatic drop in utilization of ambulatory services during the pandemic speaks to the population's fear of COVID-19 exposure. However, appropriate safeguards (social distancing, personal protective equipment, and careful training and procedure) and better communication could allow necessary care to proceed. The pandemic demonstrates the need to engage more Colombians and Venezuelans as health promotors, building on a system Colombia has operated for decades [40].

**Adjust metrics for assessing quality of care.** A further recommendation relates to the metrics used by Colombian authorities to maintain quality in service provision. According to key informant interviews in a parallel component of this study, health facilities are regularly monitored with many of the key performance indicators based on outcomes [11]. A hospital serving many migrants will probably show worse outcomes, given their restricted access to preventive services and elevated poverty and malnutrition, than a comparable hospital serving fewer migrants. This perverse incentive could discourage health facilities from serving the migrant community and providing access to non-emergency care. A more sophisticated monitoring system that adjusts for population differences (e.g. the nationality of its patients) could incentivize outreach to migrant populations.

**Create responsible units in local government.** At the department and municipal levels, governments should create units responsible for registering the displaced Venezuelans within their territories and assisting them in obtaining the relevant paperwork, registering for health insurance, and seeking employment and other services. The national government should provide training, support, and monitoring to departmental staff who, in turn, do the same for municipal staff.

**Reduce discrimination against Venezuelans.** Promising policies to promote equal access to health services include laws against discrimination in public services, creating ombudsmen and other mechanisms to investigate and redress apparent violations, and positive and empathetic communications to reduce stigma, building on a campaign initiated in 2017 [41, 42].

**Ask international donors to assume greater financial responsibility.** Finally, these findings highlight the need for international assistance in providing health services to migrants and refugees in Colombia and other countries which integrate migrants and refugees into their national health systems. Under the current system the emergency medical services provided to the unenrolled population are not covered by the insurance system. The direct financial burden is then covered by the health facility (hospital or clinic) that provided the services. Then, the facility must ask the local government's health secretariat to cover the cost. If the local government lacks sufficient resources, it may ask the national government for assistance.

However, the process is inefficient, time consuming and uncertain. In the interim, health facilities operate at a deficit, missing payments to health workers and vendors and risking collapse and sudden shutdowns. In 2018, 42% of Colombia's 930 public hospitals were at medium or high levels of financial risk [43].

Other countries with substantial numbers of refugees and displaced populations face similar challenges. UNHCR noted that in 2020 "seven in 10 people of concern to UNHCR live in urban settings, and the pandemic sharpened the challenge of supporting them" and "the needs remain vast. . .." [7] Colombia's system of national health insurance already provides a mechanism for public funding for enrollees in the subsidized regime. With international support, subsidies to Colombia's national health insurance system could be expanded to address many of these needs. These recommendations build on the authors' online knowledge brief [12].

## Supporting information

**S1 File.**
(PDF)

**S2 File.**
(PDF)

**S3 File.**
(PDF)

**S4 File.**
(PDF)

**S5 File.**
(PDF)

## Acknowledgments

The authors thank members of the Elrha project's advisory panel and speakers at the launch event for a related knowledge brief for insights, Thomas J. Bossert for thoughtful comments, Cilia Mejia Lancheros for her insightful review, and Clare L Hurley for editorial assistance.

## Author Contributions

**Conceptualization:** Donald S. Shepard, Arturo Harker Roa, Diana M. Bowser.

**Data curation:** Donald S. Shepard, Adelaida Boada, Douglas Newball-Ramirez, Carlos William Rincon Perez, Priya Agarwal-Harding, Jamie S. Jason, Arturo Harker Roa.

**Formal analysis:** Donald S. Shepard, Adelaida Boada, Anna G. Sombrio, Carlos William Rincon Perez, Priya Agarwal-Harding, Jamie S. Jason, Arturo Harker Roa, Diana M. Bowser.

**Funding acquisition:** Donald S. Shepard, Diana M. Bowser.

**Methodology:** Donald S. Shepard, Carlos William Rincon Perez, Arturo Harker Roa, Diana M. Bowser.

**Project administration:** Donald S. Shepard, Anna G. Sombrio, Arturo Harker Roa, Diana M. Bowser.

**Supervision:** Donald S. Shepard, Arturo Harker Roa, Diana M. Bowser.

**Visualization:** Adelaida Boada, Carlos William Rincon Perez.

**Writing – original draft:** Donald S. Shepard, Adelaida Boada, Arturo Harker Roa.

**Writing – review & editing:** Donald S. Shepard, Adelaida Boada, Priya Agarwal-Harding, Arturo Harker Roa, Diana M. Bowser.

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
