## [Decision Letter · Decision Letter 0]

28 Dec 2022

PONE-D-22-21282Impact of COVID-19 on healthcare utilization, cases, and deaths of citizens and displaced Venezuelans in Colombia:  Complementary components of safety net and insurance systems under a constitutional commitmentPLOS ONE

Dear Dr. Shepard,

Thank you for submitting your manuscript to PLOS ONE. After careful consideration, we feel that it has merit but does not fully meet PLOS ONE’s publication criteria as it currently stands. Therefore, we invite you to submit a revised version of the manuscript that addresses the points raised during the review process.

The manuscript tackles an important topic that deserves to be highlighted and shared. However, it looks more like a report than a research article. The authors are given a chance to improve the writing and the structure of the manuscript. The paper needs a serious revision before it is accepted. The authors are invited to address very carefully all comments provided by the two reviewers.

I also strongly recommend that authors -during their revision- follow the STROBE checklist for reporting of observational studies. Please remove subheadings from the introduction and let the research question be clearly stated at the end of the introduction: what is the aim of the study? Some information about the context in the introduction can be moved to the materials and methods under "study setting and participants". Please follow the STROBE checklist when choosing the subheadings in "Materials and Methods". On the other hand, the language needs careful and thorough revision; it is surprising that "Colombia" is often misspelled and written "Columbia". Finally, it is necessary to add a "limitations" paragraph to the discussion. 

We look forward to receiving your revised manuscript.

Kind regards,

Mabel Aoun, MD, MPH

Academic Editor

PLOS ONE

Journal Requirements:

2. Please ensure that you have specified (1) whether consent was informed and (2) what type you obtained (for instance, written or verbal, and if verbal, how it was documented and witnessed). If your study included minors, state whether you obtained consent from parents or guardians. If the need for consent was waived by the ethics committee, please include this information.

3. Thank you for submitting the above manuscript to PLOS ONE. During our internal evaluation of the manuscript, we found significant text overlap between your submission and previous work in the Introduction, Methods, Results and Discussion sections of your manuscript. We would like to make you aware that copying extracts from previous publications, especially outside the methods section, word-for-word is unacceptable. In addition, the reproduction of text from published reports has implications for the copyright that may apply to the publications. Please revise the manuscript to rephrase the duplicated text, cite your sources, and provide details as to how the current manuscript advances on previous work. Please note that further consideration is dependent on the submission of a manuscript that addresses these concerns about the overlap in text with published work. We will carefully review your manuscript upon resubmission and further consideration of the manuscript is dependent on the text overlap being addressed in full. Please ensure that your revision is thorough as failure to address the concerns to our satisfaction may result in your submission not being considered further.

“This study was primarily funded by the project "Strengthening the humanitarian response to COVID-19 in Colombia" through a grant to Brandeis University from ELRHA (https://www.elrha.org/), with DMB as PI, which supported all authors. Additional funding was provided by the World Bank (www.worldbank.org) through award WORLD 8006362 to Columbia University, Sub-award 2(PG006778-01) to Brandeis University (DSS, Brandeis PI) for the project "The big questions on forced migration in health." It supported DSS, PA-H, JSS, AHR, and DMB. The funders had no role in study design, data collection and analysis, decision to publish or preparation of the manuscript.”

Reviewers' comments:

Reviewer's Responses to Questions

**Comments to the Author**

1. Is the manuscript technically sound, and do the data support the conclusions?

Reviewer #1: Partly

Reviewer #2: Partly

2. Has the statistical analysis been performed appropriately and rigorously? 

Reviewer #1: Yes

Reviewer #2: No

3. Have the authors made all data underlying the findings in their manuscript fully available?

Reviewer #1: Yes

Reviewer #2: Yes

4. Is the manuscript presented in an intelligible fashion and written in standard English?

Reviewer #1: No

Reviewer #2: No

5. Review Comments to the Author

Reviewer #1: The authors made a significant effort to attend to points raised by the reviewers in the manuscript's previous version. The work done is methodologically correct and brings some interesting findings. However, I think that it still lacks scientific density, and contributes little to a more comphreensive discussion about healthcare access by migrants and refugees worldwide, a challenging and relevant issue. The references are still mostly reports instead of international scientific articles focused on the central thema, which would probably allow for putting the Colombian experience in a more compheensive perspective, able to contribute to building knowledge about the thema. Based in the literature, it would be important to confront the Colombian experience with the Venezuelans with other countries' experiences with migrants and refugees. Which healthcare systems' models are more responsive to the needs of migrants and refugees? What are the implications for host countries, considering different levels of resources' availability?

With regard to the writing, some problems still remain.

Reviewer #2: Thank you for the opportunity to review the paper.

This study assesses the healthcare utilization outcomes (hospitalization, consultation) and death rates during the pandemic times period (2020) and pre-pandemic period ( 2019 ) among Colombian citizens and Venezuelan people who emigrated to Colombian due to the Venezuelan political and socio-economic conditions. In addition, the author sought to identify potential disparities within and between the studied population concerning the studied outcomes.

I think the study has the potential to add important evidence on the healthcare-related condition and inequalities/ equities of Venezuelan emigrants in Colombia. However, the paper needs a better structure, and important detailed information in all sections. Also, some statements are not backed by the current study or existing data, which requires careful addressing from the authors. The text requires careful proofing/editing revisions as well. I provided my comments to the authors in attached file. Hopefully such comments could contribute to improving their manuscript.

6. PLOS authors have the option to publish the peer review history of their article (what does this mean?). If published, this will include your full peer review and any attached files.

Reviewer #1: No

Reviewer #2: **Yes: **Dr. Cilia Mejia-Lancheros

---

## [Author Response · Author response to Decision Letter 0]

14 Feb 2023

Given the large number of comments, the response is in a separate document which has been uploaded.

---

## [Editor Report · Decision Letter 1]

23 Feb 2023

Impact of COVID-19 on healthcare utilization, cases, and deaths of citizens and displaced Venezuelans in Colombia:  complementary comprehensive and safety-net systems under Colombia's  constitutional commitment

PONE-D-22-21282R1

Dear Dr. Shepard,

We’re pleased to inform you that your manuscript has been judged scientifically suitable for publication and will be formally accepted for publication once it meets all outstanding technical requirements.

Kind regards,

Mabel Aoun, MD, MPH

Academic Editor

PLOS ONE
---

## [Editor Report · Acceptance letter]

20 Mar 2023

PONE-D-22-21282R1 

Impact of COVID-19 on healthcare utilization, cases, and deaths of citizens and displaced Venezuelans in Colombia: complementary comprehensive and safety-net systems under Colombia’s constitutional commitment 

Dear Dr. Shepard:

I'm pleased to inform you that your manuscript has been deemed suitable for publication in PLOS ONE. Congratulations! Your manuscript is now with our production department. 

Kind regards, 

on behalf of

Dr. Mabel Aoun 

Academic Editor

PLOS ONE